# Exceptional increase in the creep life of magnesium rare-earth alloys due to localized bond stiffening

Deep Choudhuri [1,2], Srivilliputhur G. Srinivasan[1], Mark A. Gibson [3,4,5], Yufeng Zheng[6], David L. Jaeger[7], Hamish L. Fraser[6] & Rajarshi Banerjee[1,2,7,8]

Several recent papers report spectacular, and unexpected, order of magnitude improvement in creep life of alloys upon adding small amounts of elements like zinc. This microalloying effect raises fundamental questions regarding creep deformation mechanisms. Here, using atomic-scale characterization and first principles calculations, we attribute the 600% increase in creep life in a prototypical Mg–rare earth (RE)–Zn alloy to multiple mechanisms caused by RE–Zn bonding—stabilization of a large volume fraction of strengthening precipitates on slip planes, increase in vacancy diffusion barrier, reduction in activated cross-slip, and enhancement of covalent character and bond strength around Zn solutes along the *c*-axis of Mg. We report that increased vacancy diffusion barrier, which correlates with the observed 25% increase in interplanar bond stiffness, primarily enhances the high-temperature creep life. Thus, we demonstrate that an approach of local, randomized tailoring of bond stiffness via microalloying enhances creep performance of alloys.

[1] Department of Materials Science and Engineering, University of North Texas, Denton, TX 76201, USA. [2] Advanced Materials and Manufacturing Processes Institute, University of North Texas, Denton, TX 76207, USA. [3] CSIRO Manufacturing, Private Bag 10, Clayton South, Clayton, VIC 3169, Australia. [4] School of Aerospace, Mechanical and Manufacturing Engineering, RMIT University, Carlton, VIC 3053, Australia. [5] Department of Materials Engineering, Monash University, Clayton, VIC 3800, Australia. [6] Center for Accelerated Maturation of Materials, Department of Materials Science and Engineering, The Ohio State University, Columbus, 43210 OH, USA. [7] Materials Research Facility, University of North Texas, Denton, TX 76201, USA. [8] School of Materials Science and Engineering, Nanyang Technological University, Singapore, Singapore. Correspondence and requests for materials should be addressed to D.C. (email: deep.choudhuri@gmail.com) or to S.G.S. (email: srinivasan.srivilliputhur@unt.edu) or to R.B. (email: raj.banerjee@unt.edu)

Creep, a time-dependent inelastic deformation occurring at high homologous temperature, limits potential applications of many natural materials and engineering alloys. Traditionally, creep mechanisms in natural materials (e.g., movement of earth's mantle) and engineering alloys were inferred from rigorous experiments and served as critical inputs to creep models[1–5]. Typical mechanisms invoked by conventional creep models are non-conservative vacancy-assisted dislocation climb over obstacles, thermally activated cross-slip, solute-induced viscous drag on dislocations, jog-assisted dislocation motion, movement through dislocation intersections, and grain boundary sliding[6–19]. Such insights have helped us develop creep-resistant alloys using fine-scale precipitates and dispersion of hard particles to obstruct dislocation motion in the parent matrix[15–21].

Stress–temperature combinations that cause creep in Mg alloys during service, e.g., in automotive engine, powertrain applications for example, largely occur via dislocation climb, activated cross-slip, and/or grain boundary sliding[10–13]. Consequently, in the case of Mg alloys, the strategy for precipitation hardening typically selects alloying elements (with lower solubility in Mg, e.g., rare earth elements[10,18]) that form precipitates in high number densities on the dominant slip systems[20–22]. Coherent secondary precipitates phases were found to be particularly effective in restraining dislocation motion[18]. Furthermore, depending on processing conditions, e.g., the high-pressure

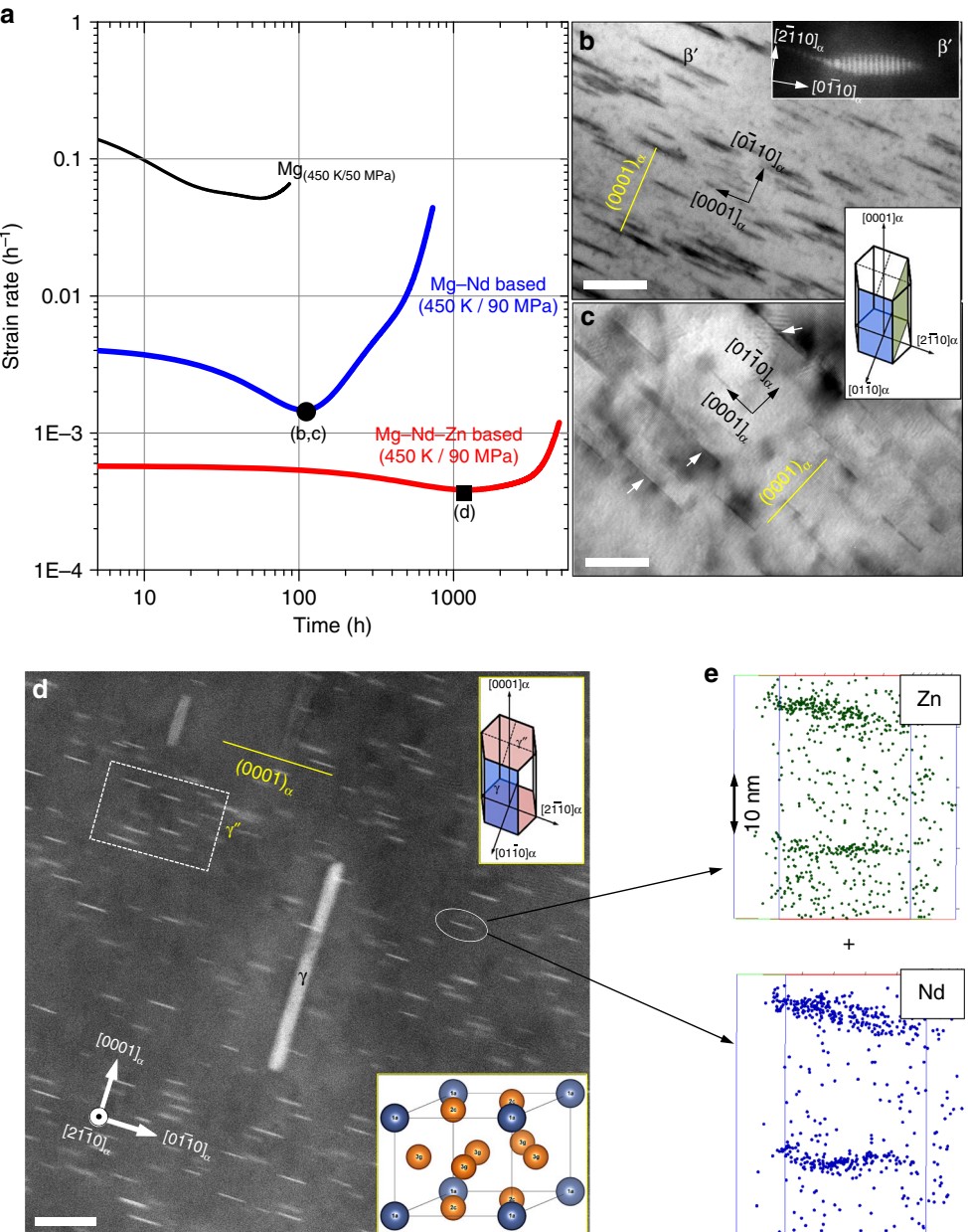

**Fig. 1** Creep response and microstructures. **a** Strain rate vs. time plots show an order of magnitude improvement in the creep life of Mg–Nd–Zn (red) over Mg–Nd (blue) and both differ significantly from Mg (black). Panels **b**–**d** are TEM and HAADF-STEM observations for Mg–Nd and Mg–Nd–Zn systems taken at points **b**–**d** marked in panel **a**. BFTEMs in panels **b** and **c** show that β′ precipitates (also inset HAADF-STEM along $[11\bar{2}0]_\alpha$) and fine-scale GP zones (arrows), respectively, in Mg–Nd lie on the prismatic planes of *hcp*-Mg. In case of Mg–Nd–Zn, HAADF-STEM in panel **d** shows γ″ and γ precipitates lying on both basal and prismatic planes, respectively. **e** Raw ion maps from atom probe tomography confirms the presence of Zn in γ″ along with Nd. Scale bars in panels **b**–**d** are 100 nm, 20 nm, and 20 nm, respectively

die-casting process, the introduction of large volume fractions of an interdendritic solidification phase/product[19] can restrict grain boundary sliding. Therefore, in the presence of such microstructural complexities, multiple mechanisms can operate in parallel during creep deformation—as reported by a recent study[21]. This underscores the need to use atomic-scale bonding considerations, in conjunction with above-mentioned conventional approach, to develop next-generation creep-resistant alloys. Our work is a significant step in this direction.

The starting point of this work is our recent exciting discovery that Zn addition to Mg–Nd–La alloy improved the creep life by 600%, which we correlated to the precipitation of large volume fraction of a new phase on the $\{0002\}_\alpha$ basal and $\{1\overline{1}00\}_\alpha/\{11\overline{2}0\}_\alpha$ prismatic planes of $hcp$-Mg[15]. While the conventional creep models that invoke inhibition of dislocation motion by precipitates in Mg–Nd–La–Zn alloys predict an increase in creep life, they may not explain the dramatic 600% increase we observed. Minor Zn addition to Mg alloys improves the creep resistance of Mg–Gd-based alloys via the formation of new Zn-containing precipitate phases[18,22,23], and segregation of Zn and Gd atoms in as-quenched bulk and at the twin boundaries[21–23]. While clearly illustrating the effects of Zn, these studies also fundamentally suggest that Zn tends to occur near rare earth (RE) atoms, which also determines the precipitation and creep response.

By correlating with atomic-level bonding, the present article sheds light on the root cause of the perplexing association between zinc and RE atoms and the improvement in creep resistance upon only minor additions[20–23]. We uncover the mechanism(s) contributing to this enhancement in creep lifetime of Mg–RE alloys by systematically correlating creep-behavior to microstructures and diffusion processes in high-pressure die-cast Mg–0.6Nd–0.4La (at%) and Mg–0.6Nd–0.4La–0.3Zn (at%) alloys (see Supplementary Tables 1 and 2), by coupling atomic-scale microstructural characterization and ab-initio simulations. Past work on Mg–La and Mg–Nd–La alloys has demonstrated that addition of La largely improves alloy castability[24], but minimally affects the creep behavior. This was attributed to the presence of La primarily inside the large interdendritic solidification phase rather than in Mg matrix[19,24,25] (also see Supplementary Fig. 1). Thus, the remainder of the article will refer to alloys with and without Zn as Mg–Nd and Mg–Nd–Zn based. Notwithstanding, we have rigorously evaluated the role of La on the creep deformation behavior using density functional theory (DFT) calculations.

Our experimental investigations determined creep–response curves of Mg, Mg–Nd, and Mg–Nd–Zn systems, and systematically characterized the creep-tested microstructures using transmission electron microscopy (TEM), including aberration corrected high angle annular dark-field scanning TEM (or HAADF-STEM) and atom probe tomography (APT). These experiments were complemented by ab-initio calculations to understand the energetics of vacancy diffusion driven creep, activated climb from the recombination of partials (related to stacking fault energy), and elastic responses. These fundamental material characteristics were further related to microalloying-induced local lattice-level pockets with covalent character (separated by regions with metallic bonding) inside the Mg matrix. Such alteration in the local lattice level bonding character was seen to appreciably enhance the local bond stiffness that resembles the modulus mismatch effect in conventional solid solution strengthening mechanism in random alloys and intermetallic phases[26–28]. However, the enhancement of creep tolerance via microalloying-induced Mg–Mg bond-stiffening effect presented here significantly differs from that in conventional solid solution strengthening mechanisms in the following manner. While our mechanism continues to operate at high homologous temperature

relevant to creep of Mg alloys, the modulus mismatch effects in conventional solid solution strengthening mechanisms precipitously deteriorate with increasing temperature. Thus, our mechanism provides a viable route to develop new creep-resistant materials.

## Results

We have studied the creep-response and the associated microstructure using multiple techniques. First, we show how microstructures in Mg, Mg–Nd, and Mg–Nd–Zn correlate with their creep behavior presented in strain rate time plots in Fig. 1a. We define creep lifetime as the time for the onset of tertiary creep (marked with arrows in Fig. 1a). It is seen that Nd addition significantly delays the onset of tertiary creep from ~60 h for the pure Mg reference material to 460 h for the base alloy. Remarkably, a further addition of Zn to Mg–Nd alloy extends the onset time to ~3300 h, and reduces the minimum creep strain rate by an order of magnitude (also see Supplementary Fig. 1). Furthermore, we emphasize that Mg–Nd–Zn alloy did not fail even after ~4800 h of testing. This behavior was attributed originally to the differences in the precipitate distribution in these two alloys[20]. To understand this significant improvement in creep performance by microalloying, we have investigated how the crystallographic orientation of precipitates and their distribution influence the creep behavior by comparing the Mg–Nd and Mg–Nd–Zn alloy microstructures at the minimum creep rate. Figures 1b and c show bright-field TEM (BFTEM) images of two types of microstructural entities near the $[11\overline{2}0]_\alpha$ zone of the $hcp$-Mg matrix (hereon referred to as α). Mg–Nd primarily contained coarse (~20–100 nm long) lenticular β′ (orthorhombic structure[18], also see the Supplementary Note 2) precipitates (inset in Fig. 1b) and fine-scale (~10–20 nm long) GP zones (Fig. 1c)—consistent with literature reports[18]. These precipitates formed primarily on the $\{11\overline{2}0\}_\alpha$ and $\{01\overline{1}0\}_\alpha$ prismatic planes of the α-Mg matrix as shown schematically in the middle inset figure. In contrast, precipitates in Mg–Nd–Zn form on both basal $(0002)_\alpha$ and prismatic planes of α-Mg. Figure 1d shows a HAADF-STEM image of the precipitates recorded with the electron beam approximately parallel to $[11\overline{2}0]_\alpha$, and the top-right inset figure shows their habit plane. The highest number density of fine precipitates occurred on the basal planes (see Supplementary Table 3), with number densities an order of magnitude higher than those observed in case of binary Mg–Nd, whilst a smaller volume fraction of coarser plate-like precipitates are contained in the prismatic planes. The fine and coarse precipitates are γ″ and γ, respectively[18,20]. The γ″ has a hexagonal structure with space group $P\overline{6}2m$[18,20] and γ has a cubic structure with space group $Fm\overline{3}m$, like β_1 phase in the Mg–Nd alloy (see Supplementary Note 2)[18,20]. APT analysis of these γ″ precipitates clearly revealed an enrichment of Nd and Zn within them (shown in Fig. 1e). Furthermore, at the minimum creep rate, the number density of γ″ in Mg–Nd–Zn is one order of magnitude larger than β_1 precipitates in Mg–Nd[20]. Therefore, from this comparative microstructural analysis and creep testing results, we infer that γ″ precipitates on the basal planes must play a crucial role in enhancing the creep strength of Mg–Nd–Zn. A description of precipitation sequence in Mg alloys is included in the Supplementary Note 2.

The structure and stability of the critical strengthening precipitate γ″ has been investigated here in detail, with a focus on the most probable site/elemental occupancies in this ternary intermetallic compound consisting of Mg as the primary element with varying lattice site occupancies of Zn and Nd. The γ″ differs from the simpler binary phases typically observed in the Mg–Nd system, and reported in the literature. These differences have been explained in detail, based on our experimentally observed and

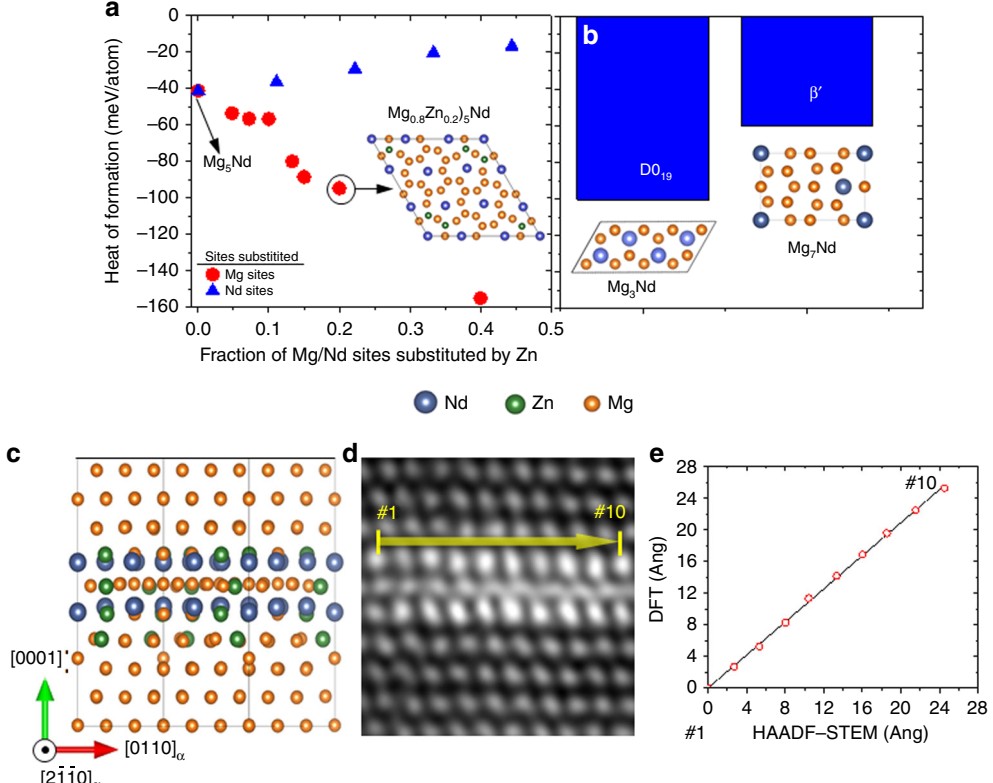

**Fig. 2** γ″ structure after Zn substitution. **a** Heat of formation of γ″ is plotted as a function of the fraction of Mg and Nd sites substituted by Zn. **b** Heats of formation for D019 and β′. Panels **c–e** show experimental observation and DFT calculations agree well: **c** DFT-derived structure of $(Mg_{0.8}Zn_{0.2})_5Nd$ γ″ supercell sandwiched in a Mg matrix, **d** atomic resolution HAADF-STEM image of γ″, and **e** plot comparing interatomic distances obtained from DFT calculations and measurements performed in the HAADF-STEM image

simulated electron diffraction patterns, based on ab-initio calculations, and shown in Supplementary Fig. 2a–d. The basic unit cell of γ″ with Wyckoff positions Nd1 *1a*, Mg1 *3g*, and Mg2 *2c* (stoichiometry $Mg_5Nd$ and space group $P\bar{6}$ 2m) is shown in the bottom inset of Fig 1d[18]. Subsequently, γ″ supercells with differing site occupancies were constructed by substituting either Mg or Nd sites with Zn atoms in varying fractions, whilst maintaining the symmetry.

The stability of γ″, D0₁₉ (stoichiometry $Mg_3Nd$), and β′ (stoichiometry $Mg_7Nd$) phases were evaluated at 0 K (Fig. 2a), by calculating their heats of formation ($H_f$). These results, in conjunction with APT and HAADF-STEM results, were used to examine the role and stability of precipitate matrix interfaces. The D0₁₉ structure is believed to be present in Mg–Nd as GP zones[18,29]. We found that energetics favor Zn substitution at Mg sites over Nd sites. Interestingly, one of the predicted γ″ compositions (Mg–16Nd–16Zn (at.%) or $(Mg_{0.8}Zn_{0.2})_5Nd$ and marked in the inset of Fig. 2a) is comparable to the γ″ composition determined from APT: Mg–(13–14at.%)Nd–(9–13at.%)Zn. The APT compositions were determined by employing a cluster analysis algorithm to several APT data sets[30,31]. Additionally, γ″–$(Mg_{0.8}Zn_{0.2})_5Nd$, with $H_f$ of −95 meV/atom and hereon referred to as γ″, has an energetic stability similar to D0₁₉ (−100 meV/atom). However, it is more stable than β′ (−60 meV/atom). From these values, we can infer that the formation of D0₁₉ and γ″ phases, in Mg–Nd and Mg–Nd–Zn respectively, are equally favored. Yet, phases that are stable at T = 0 K may not be so at higher temperatures, and may even be absent in the microstructure. For example, the bulk D0₁₉ structure, despite its large $H_f$ value, does not exist as a monolithic precipitate within the *hcp*-Mg matrix[32,33]. Instead, GP zones

are reported to exist as very fine nano-scale ordered arrangements within the *hcp*-Mg matrix and not with the postulated D0₁₉ structure[33]. This indicates that the energy of the precipitate–matrix interface is likely to play a significant role in stabilizing the precipitate.

A detailed study of γ″/Mg interface is beyond the scope of this work. However, our TEM studies of Mg–Nd–Zn microstructure suggests that a stable γ″/Mg interface exists, unlike unstable D0₁₉ in Mg–Nd. The structure of γ″–$(Mg_{0.8}Zn_{0.2})_5Nd$ was further confirmed by comparing our DFT calculations with atomically resolved aberration corrected HAADF-STEM microscopy results presented in Fig. 2c and d, respectively. The HAADF-STEM images, shown in Fig. 2d, and in Supplementary Fig. 3, clearly reveal higher intensity atomic columns, corresponding to heavier elements[34,35], at the same locations where the Nd and Zn atoms occupy the γ″ structure in the DFT simulations (Fig. 2c). Additionally, the distances between the atomic columns along the row exhibiting the brightest intensity, marked as #1 to #10, are in excellent agreement with those obtained from the DFT-simulated structure, as shown in Fig. 2e. Such a coupled TEM–DFT approach was essential because 2D projections associated with TEM, irrespective of resolution, do not reliably allow identification of Zn site in the γ″ lattice. The large $H_f$ of γ″ when Zn substitutes in Mg–Nd, and its high number density, and widespread presence and stability (does not dissolve; see Supplementary Fig. 4), after prolonged creep testing at elevated temperature, leads us to hypothesize that the γ″ precipitates are likely to be the critical barrier to dislocation glide during creep at higher stresses. We will also emphasize that the precipitates in both alloys will coarsen with prolonged creep (see Supplementary

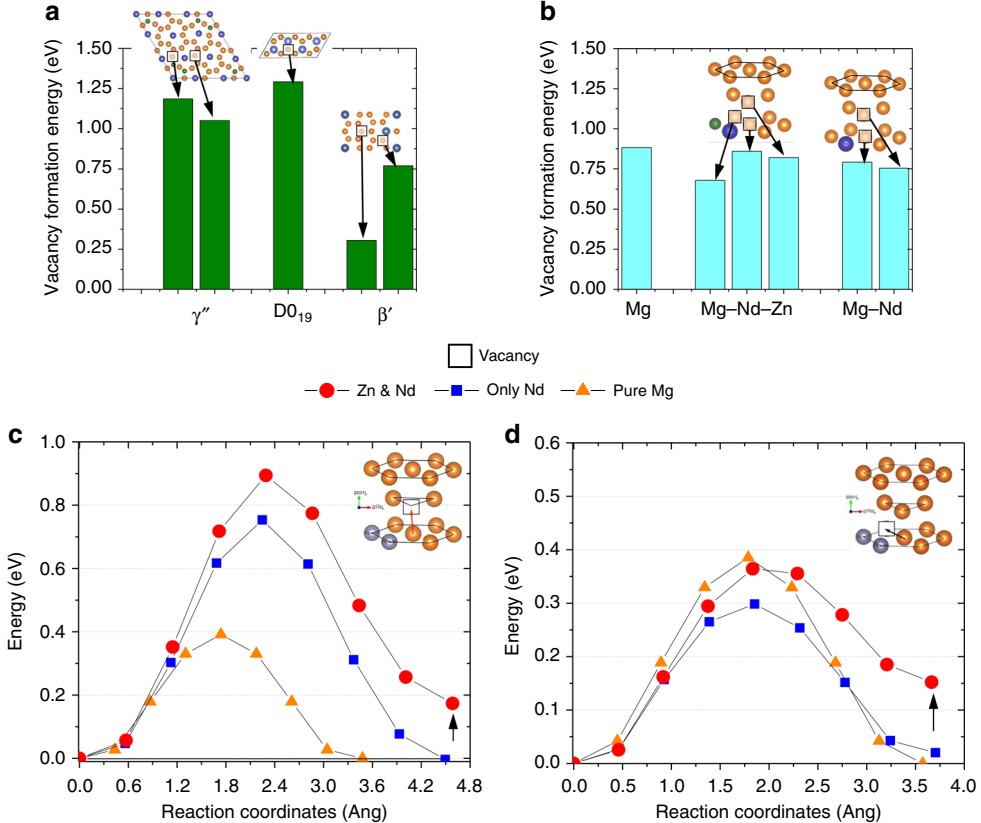

**Fig. 3** Formation and migration energies of vacancy at Mg sites. Heat of vacancy formation at Mg sites in: **a** γ″, D0₁₉, and β′ structures; **b** hcp-Mg lattice with no solutes, Nd and Zn solutes, and only Nd solute. Energies were calculated for vacancies at different nearest neighbor distances to Nd and/or Zn. Energy vs. reaction coordinates plots showing transition state calculations for out-of-plane (**c**) and in-plane (**d**) vacancy migration. The insets in **c** and **d** schematically show the migration paths in hcp-Mg. Addition of Zn retards Mg vacancy migration by increasing the peak-barrier and final saddle point energies (marked with arrow in **c**–**d**)

Fig. 4 and ref. [36]), and that will inevitably reduce creep lifetime of both systems.

Presently, in order to explain an order of magnitude reduction in the minimum creep rates, we further used precipitate number densities in Mg–Nd and Mg–Nd–Zn, measured from BFTEM images, as inputs to different strengthening models to estimate creep enhancements in the Mg–Nd–Zn system (see Supplementary Note 3). Furthermore, our DFT calculations revealed that elastic moduli of γ″ are comparable to pure Mg (see Supplementary Table 4 and Supplementary Note 3) and hence do not significantly strengthen Mg matrix by the modulus effect[14,18,37,38]. Classical precipitation strengthening models[14,37,38] predict only a 3-fold increase in the critical resolved shear stress (CRSS) for the γ″ fractions estimated in Mg–Zn–Nd (see Supplementary Table 5). However, any interpretation solely based on microstructure fails to explain the observed improvement in the creep lifetime in Mg–Nd–Zn. Reasonable explanations must also consider vacancy interaction with precipitates and solutes, and acceleration of dislocation climb over impeding precipitates, driven by enhanced vacancy diffusion at high homologous temperature, and activated cross-slip of screw dislocations[10–13].

We now determine vacancy energetics using first-principle calculations to understand their role in the superior creep lifetime of our Mg–Nd–Zn alloy. The formation energies of Mg vacancies in γ″, D0₁₉, and β′ precipitates are compared with that in the hcp-Mg lattice with different solute environments (Fig. 3b). Figure 3a and b show formation energy ($E_f$) of Mg vacancy sites that are at different nearest neighbor (NN) distances from

Nd, and Nd and Zn. They reveal that vacancies form more easily in pure Mg and without solutes ($E_f = 0.68$–$0.88$ eV) than in γ″ ($E_f = 1.19$ and $1.05$ eV) and D0₁₉ ($E_f = 1.29$ eV) precipitates. Surprisingly, vacancy formation energy values of $E_f = 0.30$–$0.77$ eV in the coarser β′ precipitates are comparable or less than those in the Mg lattice. Thus, vacancies will easily form in Mg–Nd, both in Mg matrix and in β′—in potentially higher concentrations. They will undoubtedly be an important facilitator of dislocation climb during creep. In contrast, the larger vacancy formation energy in γ″ precipitates, present in larger densities in Mg–Nd–Zn than either D0₁₉ or β′ in Mg–Nd, suggests that vacancy formation in γ″ is less likely. Instead vacancies will form predominantly within the Mg solid solution, and that vacancy migration within the Mg–Nd–Zn alloy matrix may likely play a key role in creep.

In hcp metals, there are two primary paths for vacancy diffusion—out-of-plane (OOP) between two (0002)ₐ planes and in-plane (IP) within a (0002)ₐ plane[39]. Thus, we next calculated migration energies for these paths in Mg, Mg–Nd, and Mg–Nd–Zn matrices via the nudged-elastic-band (NEB) method[40], using 96 atom supercells for numerical accuracy (see Supplementary Fig. 5 for the vacancy diffusion paths). The OOP and IP migration barriers for reference pure Mg are 0.38 and 0.39 eV, respectively, and in agreement with literature[41]. Figure 3c and d plot minimum energy path (MEP) for OOP and IP vacancy diffusion for different solute environments (also see extended Fig. e4). The addition of a Zn atom as the first neighbor to an Nd atom substituted within the Mg lattice increased the OOP migration energy ~125% compared to the

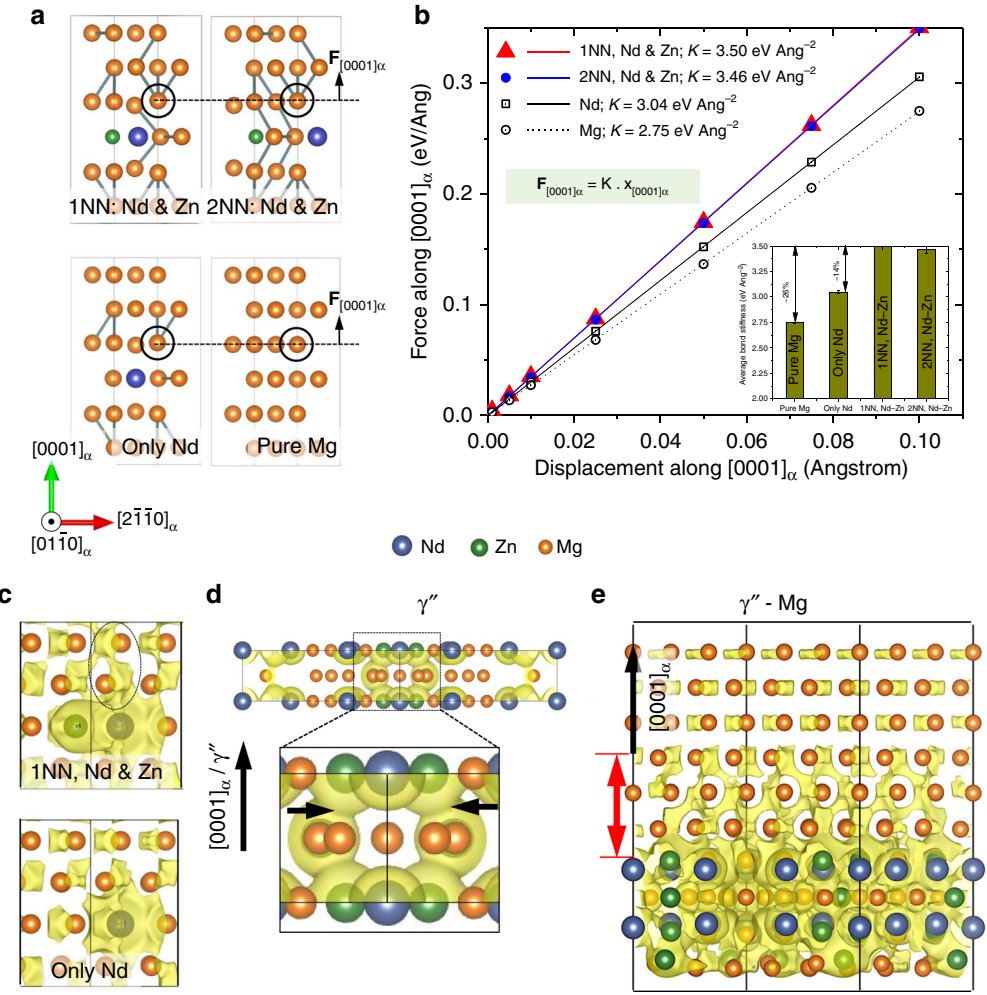

**Fig. 4** Bonding character from DFT calculations. **a** Mg lattice supercells showing Nd and Zn solutes as 1st and 2nd nearest neighbors (NN), only Nd solute, and pure Mg. The encircled Mg atom was displaced along the $[0001]_\alpha$ to calculate the restoring force. **b** Plot of DFT-calculated force as function of displacement along $[0001]_\alpha$. The inset histogram shows the calculated bond stiffness increases with Zn addition. Charge isosurface using $\Delta\rho = 0.015$ eÅ$^{-3}$ in panels **c–e** shows the valence electron delocalization along $[0001]_\alpha$ in solid solution, bulk $\gamma''$, and $\gamma''$-Mg supercells in Mg–Nd–Zn system

pure Mg (Fig. 3c), and ~18% with respect to only Nd substitution. The Zn addition, on the other hand, did not appreciably change IP vacancy migration barrier for Mg diffusion (Fig. 3d). From Fig. 3c and d, we observe that one of the saddle points for Mg–Nd–Zn (marked with arrow) has a higher energy than the initial configuration. This causes the OOP and IP vacancy migration in the presence of both Zn and Nd to create an energetically unfavorable configuration. Furthermore, as described above and summarized in Fig. 3c and d, the OOP and IP vacancy migration barrier energies follow the trend: $E^{\text{OOP}}_{\text{Mg–Nd–Zn}} > E^{\text{OOP}}_{\text{Mg–Nd}} > E^{\text{OOP}}_{\text{Mg}}$, and $E^{\text{ip}}_{\text{Mg–Nd–Zn}} \approx E^{\text{ip}}_{\text{Mg}} > E^{\text{ip}}_{\text{Mg–Nd}}$. Interestingly, this trend in OOP energy barriers correlates well with the increase in creep strength and failure-time in Fig. 1a. NEB calculations involving Mg–La and Mg–La–Zn also indicated similar trends (Supplementary Fig. 6 and Supplementary Note 5). Thus, we can infer that OOP vacancy diffusion, necessary for dislocation climb, appears to be the rate-determining step of creep in Mg–Nd–Zn alloys.

The trend in vacancy migration barriers was correlated to $\langle 0001 \rangle_\alpha$ and $\langle 11\bar{2}0 \rangle_\alpha$, the Mg–Mg bond stiffness ($K$) by varying the solute environments (Fig. 4a). The bond stiffness was determined systematically by displacing a Mg atom along $[0001]_\alpha$ and determining the corresponding restoring Hellman–Feynman forces (plotted in Fig. 4b). The excellent straight-line fits in Fig. 4b

indicated that the displacements were within harmonic limits and allowed us to calculate $K$ for each solute environment from the linear slopes. The results show that the presence of Zn and Nd (as 1st or 2nd NN) increased $[0001]_\alpha$ stiffness by 26% and 14% compared to Mg and Mg–Nd systems. The bond stiffness along the $a$-axis ($\langle 11\bar{2}0 \rangle_\alpha$) did not differ significantly between pure Mg and the Zn-containing supercells (Supplementary Fig. 7). Importantly, bond stiffness trend parallels the OOP migration barriers: $K^{\text{1NN}}_{\text{Mg–Nd–Zn}} \approx K^{\text{2NN}}_{\text{Mg–Nd–Zn}} > K_{\text{Mg–Nd}} > K_{\text{Mg}}$.

The higher bond stiffness along $[0001]_\alpha$ in Mg–Nd–Zn, and its correlation with OOP diffusion barrier, was further analyzed using the electronic charge distribution around the solute atoms in the Mg lattice (Fig. 4c). The presence of Zn near an Nd atom localizes anisotropically the excess electron density along $[0001]_\alpha$ as shown by dotted ellipses around Zn and Nd (Fig. 4e). Such valence electron density localization was also verified in larger 500 supercells where Nd and Zn concentrations are well within their experimentally measured values. This localization signifies an increased covalent bond character in Mg–Nd–Zn[27,28], which correlates with the increased stiffness along $[0001]_\alpha$. Such Zn-addition-induced covalent character also exists in other cases. Electron charge localization between the $(0001)_{\gamma''}$ binds the two planes (shown by arrows in Fig. 4d) and stabilizes $\gamma''$ as evidenced by its more negative $H_f$ (Fig. 2a). In contrast, such interplanar

localization is absent in $\gamma''$ precipitates without Zn substitution and when Zn substitutes for Nd atoms (Supplementary Fig. 8a), which also explains the higher formation energies of these structures (Fig. 2a).

Electronic charge distribution around the $\gamma''$/Mg interfacial regions was also studied (Fig. 4d). The calculations showed that electronic charge were localized at three $(0002)_\alpha$ planes of the Mg lattice in $\gamma''$/Mg interface, while charge localization persisted to the nearest $(0002)_\alpha$ plane of the Mg–Nd–based structures (Supplementary Fig. 8b). This observation is significant because the $\gamma''$ precipitates are better bonded to its surrounding parent Mg lattice than $\beta'$ in Mg–Nd, and the enhanced covalent bonding of the $\gamma''$/Mg interfacial regions can likely inhibit diffusing vacancies during creep. Additionally, our recent study of precipitate–matrix interfaces in a Mg–RE alloy indicated that lower interfacial energy correlates with higher interfacial valence charge density (also see Supplementary Note 3)[42]. Taken together, the stiffer bonds near the solute species and $\gamma''$/Mg interfacial regions will form a distribution of covalently bonded pockets within the matrix of Mg–Nd–Zn. These stiff pockets will raise the vacancy migration barrier in Mg–Nd–Zn by an order of magnitude over Mg–Nd, and correspondingly reduces the dislocation-climb-dominated creep rates by the same factor (Supplementary Note 4 and Supplementary Tables 6 and 7).

Finally, to evaluate the role of activated cross-slip during creep, basal stacking fault energy (SFE) of 1/3 $\langle 1\bar{1}00 \rangle_\alpha$ partial dislocations in pure Mg, Mg–Nd, and Mg–Nd–Zn alloys were computed from their generalized stacking fault energy (GSFE) curves (see Supplementary Methods Section). In Fig. 5a, the SFE for the intrinsic I2 stacking faults in hcp metals[43,44] corresponds to the first minima of the GSFE curves, and they vary as $I2_{Mg–Nd–Zn} < I2_{Mg–Nd} < I2_{pure\ Mg}$. Note that $I2_{Mg–Zn} \approx I2_{pure\ Mg}$ (see Supplementary Fig. 9). For partials (with screw character[14]) to cross-slip to a different plane, they first need to recombine into a perfect lattice dislocation, and the work done to drive that process is equivalent to the dislocation dissociation energy of the lattice dislocation. The SFE was related to the dissociation energy (see equation 2 in the Supplementary Methods Section). The calculated values are plotted as bars in Fig. 5b, which also compares the dissociation energies of pure Mg, Mg–Nd, and Mg–Nd–Zn alloys with their OOP and IP migration barriers (Fig. 3c, d). We note that the addition of Zn to Mg–Nd increases the dissociation energy, which means that Zn is more effective in hindering activated cross-slip in Mg–Nd–Zn alloys. Furthermore, the calculated higher OOP migration energy (see Fig. 5b) demonstrates that OOP dislocation climb faces a greater energy penalty than IP vacancy diffusion and activated cross-slip. Similar results were also obtained for Mg–La and Mg–La–Zn (Supplementary Note 5 and Supplementary Fig. 9).

## Discussion

This letter has identified several key mechanisms responsible for the dramatic improvement of creep lifetime of Mg–Nd–Zn. First, the presence of Zn-stabilized $\gamma''$ on basal planes and the cubic $\gamma$ precipitates on the prismatic planes (Fig. 1d) restricts dislocation movement on the dominant slip systems of Mg–Nd–Zn compared to arresting only the prismatic slip in Mg–Nd. This physical encompassing of the hcp-Mg lattice via precipitates has been postulated earlier from "honeycomb" structures seen in Mg–Nd–Y alloys,[18]. However, with Zn addition, we have been able to attain a very high number density of $\gamma''$ and precipitation on both basal and prismatic planes. Regardless, conventional creep models show that the $\gamma''$ precipitate orientation and their high number density only enhances the CRSS of Mg–Nd–Zn by a factor of three. Second, we show that the presence of both Zn and

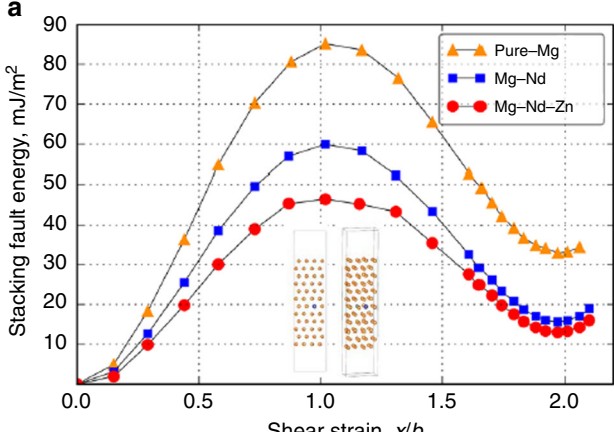

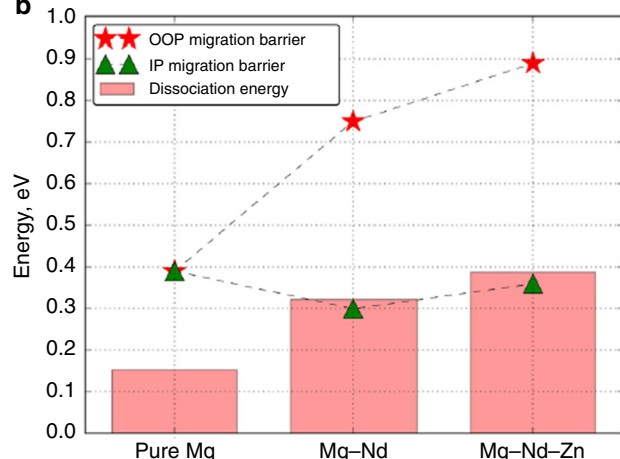

**Fig. 5** Generalized stacking fault (GSF) and dislocation dissociation energies. **a** Plot depicting the variation of fault energy with planar displacement along $\langle 11\bar{2}0 \rangle_\alpha$ in pure Mg, Mg–Nd, and Mg–Nd–Zn. Bottom inset figures show two views of the orthogonal cell used in the calculations. Stacking fault energy corresponds to the first minima of the GSF energy curves. Addition of Zn to Mg–Nd further reduces SFE by ~17%. **b** Histogram comparing the energy required to dissociate a perfect (1/2) $\langle 11\bar{2}0 \rangle_\alpha$ dislocation into two (1/3) $\langle 1\bar{1}00 \rangle_\alpha$ partials for pure Mg, Mg–Nd, and Mg–Nd–Zn alloys. Adding Zn to Mg–Nd increases the dissociation energy by ~20%. Out-of-plane vacancy migration energies, also indicated in the same plot, are also significantly larger than the dissociation energies

Nd strengthens the interplanar bonding by imparting covalent bond character within the Mg matrix and at the $\gamma''$/Mg interfaces. A distribution of such local covalently bonded regions inside the Mg matrix of Mg–Nd–Zn alloy increased the OOP vacancy migration energy barrier by an order of magnitude compared to such a barrier in Mg–Nd. Third, the increased vacancy migration barrier correlated with an order of magnitude reduction in creep strain-rate of Mg–Nd–Zn. Therefore, the alteration of electronic structure via alloying with Zn is a potential approach for alloy strengthening because we are fundamentally imparting covalent character to increase the bond strength. Finally, Zn addition also likely hinders the activated cross-slip by reducing the stacking fault energies. Thus, we note that the creep response of the Mg–Nd–Zn alloy is enhanced by the interaction of multiple factors related to chemical bonding along the $[0001]_\alpha$ crystallographic axis of hcp-Mg. However, it needs to be determined whether these mechanisms affect the creep strength in either linear or non-linear combinations.

In conclusion, using Mg–Nd and Mg–Nd–Zn model alloys, we have discovered that engineering stiffer directional bonds by using appropriate microalloying additions can dramatically improve their high temperature creep response. Our findings have bearing not only on the design of ultra-lightweight creep resistant Mg alloys, but may also guide the future development of other alloy systems.

## Methods

Alloys of composition (in at%) Mg–0.6Nd–0.4La and Mg–0.6Nd–0.4La–0.3Zn were creep tested at 450 K with a constant tensile stress of 90 MPa. Characterization of the deformed microstructure was performed with FEI's Tecnai TF20 and Titan3 80–300 TEMs, while APT was performed in a LEAP 3000X HR. DFT calculations were performed with Vienna Ab-inito Simulation Package (VASP) using the projector augmented wave method within the generalized gradient approximation of Perdew, Burke and Ernzerhof[45,46]. MedeA software was used for generating Zn-substituted precipitate structures, calculation of elastic constants, and computing/analyzing charge density distributions[47]. All calculations were carried out with a cutoff voltage of 350 eV and dense k-point sampling to ensure numerical accuracy. Additional details are presented in the Supplementary Methods Section.

**Data availability**. All data are available from the authors.

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

## Acknowledgements

The computations were done on UNT's Talon3 and Texas Advanced Computing Center's Stampede supercomputer. The authors used the Materials Research Facility at UNT. D.C., R.B., and S.G.S. acknowledge support from NSF grant CMMI-1435611, and H.L.F. and Y.Z. were supported by NSF grant CMMI-1435483. Dr. T. Alam helped us with atom probe experiments.

## Author contributions

D.C., S.G.S., M.A.G., H.L.F., and R.B. contributed in designing the experiments. M.A.G. prepared the alloys used in this study and performed the creep experiments. D.C. and S. G.S. designed and executed the simulations. D.C., S.G.S., M.A.G., and R.B. wrote the manuscript with inputs/discussion with other authors. D.C. performed conventional TEM examinations and analysis of the results. Y.Z. recorded HAADF-STEM images and D.L.J. acquired and analyzed atom probe data.

## Additional information

**Competing interests:** The authors declare no competing financial interests.

