## [Peer Review File · Nature Communications]

Reviewers' Comments:

Reviewer #1:

Remarks to the Author:

The authors have attempted to explain the role of Zn in a Mg alloy containing 3.4wt% Nd and 2.2wt%La on its creep behavior via first principles calculations. A very commendable attempt, however, in its present framework, the manuscript is not acceptable for publication.

The points raised below are not in order of importance but is a partial list of concerns in the sequence they appear in the manuscript.

Lines 22-23: No references are given as to which Mg alloys benefit from "small additions of amountsof elements such as Zn" with respect to creep resistance. Without the references it is hard to assess the possible role of Zn on creep resistance. It should also be noted that there is nothing puzzling about this fact as the authors mention in the next sentence.

Line 25: Compositions of the Mg-Nd and Mg-Nd -Zn alloys are not given in the text or in the summary but appear only in the additional section at the end. First, it is very frustrating to read about an alloy without its composition and secondly only at the end we discover that the alloys also have La. La is not discussed or studied at all in this work.

Lines 39-42: The description of creep in this section is incomplete which is surprising when there are so many reviews on the mechanisms of creep in Mg alloys. Dislocation glide on active slip systems is not a proper creep mechanism, it occurs only at very high stresses close to the yield strength of the alloys at the creep temperature. Dislocation climb is also only one of the creep mechanisms among others (eg. activated cross slip, movement of dislocations with solute atmospheres, grain boundary diffusion or grain boundary slide and many others as the temperature increases).

Lines 45: It is well known that precipitation hardening with metastable phases is not a way to increase creep resistance as these precipitates would coarsen or transform to other metastable or stable phases during creep. Age hardening precipitates are rarely stable or even quasi stable at creep temperatures. It is noted that the authors have not shown TEM images from the tertiary creep stage (Fig. 1), where the creep rate increases; as such we cannot rule out precipitate coarsening and transformation as the main reason for the loss of creep resistance at longer life.

Lines 55 onwards: The simple way to determine the creep mechanism in an alloy is by generating Arrhenius plots with multiple temperatures with constant stress or multiple stresses at constant temperature to obtain the activation energy or the stress exponent. 600% or orders of magnitude increase in creep is very common in Mg alloys and these are of course not explained by mechanisms related to increase of strength which is quite different from creep resistance. The creep resistance of the alloys are not related to their strength or even to their high temperature strength because for the former, time is not a factor and the stresses involved are athermal versus thermally activated. Lines 177-183: The argument in this section that the precipitates can only strengthen 3-fold does not at all apply to creep and hence the argument and the claim in italics are not acceptable.

Line 59-66: Again in this section none of the alloy composition limits are given. We finally see in the additional section that the alloy has 0.8wt% Zn. Well Zn has about 2wt % binary solubility in Mg at room temperature which increases towards 6wt% at high temperatures. But the alloys also have 3.4wt%Nd and 2.2wt%La. The authors should give the solubility limits of each element in alpha-Mg matrix of these alloys. Without this information the reader does not know if Zn is in solid solution (if so how much) and how much is in the precipitates. Nd on the other hand has no binary solubility but again it is not known if in the multi-component alloy the solubility limits change. Why is this important? Studies have indicated that Zn in solid solution is unusually surface active in Mg which is surprising because the Wigner Seitz radii do not predict that Zn would be surface active in Mg. It has been observed that Zn actually segregates to grain boundaries and stays there tenaciously and does not easily homogenize into the Mg grain even when soaked at high temperatures. If so it is highly likely that Zn would also segregate to dislocations and create

dislocation atmospheres which are not easily lost at creep temperatures; this could explain the increase in creep resistance when Zn is added. It is also likely that Zn would stick to other defects and interfaces in the alloy; Zn could enrich at the outer surface of the precipitates and increase their stability or change their habit plane. If Zn interacts with vacancies it would also make it difficult for the GP zones to form, which need vacancies (unless if Zn itself acts as nuclei for the GPs)

Line 206: The DFT calculations on vacancy migration pathways are interesting. Again it is not clear what composition range is simulated with the 96 atom supercell. In creep deformation there are many other pathways for vacancy migration (grain boundaries, fault boundaries, dislocations). It is not clear how the vacancy barrier has been related to the creep rates in Fig. 1 without an indication of the operative creep mechanism. It is also not shown how Nd and La would influence these pathways.

Line 245: very difficult to accept covalent bonding in the Mg matrix unless if the DFT is actually looking at the composition ranges where intermetallics exist. The increase of covalent bonding in the intermetallics may have some bearing for creep resistance. Increased covalent bonding would lead to a less ductile intermetallic; this however can have both beneficial and negative effects on creep resistance.

The authors should either address the points mentioned above and improve /complete the analysis and the background on creep , or prepare the manuscript as a DFT study of the interaction of Zn with the precipitates and two vacancy pathways in Mg-0.6Nd-0.4La (at %) alloys.

Reviewer #2:

Remarks to the Author:

In this work, increased vacancy diffusion barrier is shown to correlate with 25% increase in inter-planar bond stiffness and is identified as the key mechanism in the enhancement in high-temperature creep strength.

Overall, through an integrated first-principles modeling and high-resolution TEM and APT characterization study, the "localized bond stiffening" via micro-alloying is demonstrated as a viable route to enhance creep strength (by up to 600%).

The findings are novel, the approach is rigorous using state-of-the-art capabilities and work represents a significant new advance in the field of precipitation-hardened Mg-alloys.

Minor correction:

Change "complimented" to "complemented" on pg 3

Response to Reviewer #1 comments

The authors thank the referee for his careful review, helpful comments, and service. The extremely helpful comments have motivated several new calculations and additional analysis that have yielded new insights. Below we have addressed each of the reviewer's comments in detail, and highlighted the corresponding changes we made in the manuscript.

Comment 1: Line 25: Compositions of the Mg-Nd and Mg-Nd –Zn alloys are not given in the text or in the summary but appear only in the additional section at the end

Response: The authors apologize for this oversight and have added this in the revised manuscript.

Manuscript introductory section:

“We uncover the mechanism(s) contributing to this enhancement in creep life-time of Mg-RE alloys by systematically correlating creep-behavior to microstructures and diffusion processes in high pressure die cast Mg-0.6Nd-0.4La (at%) and Mg-0.6Nd-0.4La-0.3Zn (at%) alloys (see Tables T1 and T2 in the supplementary section), by coupling atomic scale microstructural characterization and ab-initio simulations.”

Manuscript Supplementary section:

Table T1. Compositions of Mg-Nd and Mg-Nd-Zn alloys measured with inductively coupled plasma atomic emission spectroscopy

Alloy	Nd wt%	La wt%	Zn wt%	Ce wt%	Gd wt%	Pr wt%	Y wt%	Al wt%	Fe	Be	Cu	Ni
Mg- Nd	3.50	2.49	0.00 5	0.05	0.01	<0.0 1	<0.0 1	0.02 7	0.00 5	<0.0 01	<0.0 01	<0.0 01
Mg- Nd- Zn	3.3	2.43	0.77	0.04	<0.0 1	<0.0 1	<0.0 1	0.00 8	61 ppm	-	<1pp m	<1pp m

Comment 2: Lines 39-42: The description of creep in this section is incomplete which is surprising when there are so many reviews on the mechanisms of creep in Mg alloys. Dislocation glide on active slip systems is not a proper creep mechanism, it occurs only at very high stresses close to the yield strength of the alloys at the creep temperature. Dislocation climb is also only one of the creep mechanisms among others (eg. activated cross slip, movement of dislocations with solute atmospheres, grain boundary diffusion or grain boundary slide and many others as the temperature increases).

Response: We were constrained from addressing this aspect in the original manuscript because of space limitations. However, based on the above comment we have added the following:

In the Introductory Section:

“The conventional creep models invoke the following mechanisms: (i) non-conservative vacancy assisted dislocation climb over obstacles, (ii) thermally activated cross-slip, (iii) solute-induced viscous drag on dislocations, (iv) jog-assisted dislocation motion, (v) movement through dislocation intersections, and (vi) grain boundary sliding [6-19].”

“Stress-temperature combinations that cause creep in Mg alloys during service, e.g. in automotive engine, powertrain applications for example, largely occur via dislocation climb, activated cross slip, and/or grain boundary sliding [10-13].”, and

“Furthermore, depending on processing conditions, e.g. the high-pressure die-casting process, the introduction of large volume fractions of an interdendritic solidification phase/product [19] which can restrict grain boundary sliding. Therefore, in the presence of such microstructural complexities, multiple mechanisms can operate in parallel during creep deformation – as reported by a recent study [21].”

We now describe how we have estimated the contribution of various mechanisms to the creep behavior in our alloy.

Activated cross-slip: The cross-slip mechanism first involves the recombination of the Shockley partial dislocations into a single screw lattice dislocation, which subsequently splits into partials on the cross-slip plane to complete the cross-slip process. The energy required to recombine the partials is defined as “*dislocation dissociation energy.*” Widely split dislocations in low stacking fault energy (SFE) metals need more energy to recombine and hardly cross-slip. The contribution of cross-slip mechanism to creep was assessed by calculating the generalized stacking fault energy (GSFE) curves using DFT, which enable us to calculate SFE and dissociation energies of lattice dislocations. These new calculations were performed for Pure-Mg, Mg-Nd-Mg-Nd-Zn, Mg-La and Mg-La-Zn. Please refer below for the results on Mg-La and Mg-La-Zn results.

During creep deformation, the thermal energy need to be similar to *dislocation dissociation energy* to initiate activated cross-slip. We found that dislocation dissociation energy in Mg-Nd-Zn is 20% larger than in Mg-Nd alloys, making it harder for cross-slip of screw dislocations in the former.

Accordingly, the detailed methodology is given in the supplementary section and the main results are included in the text.

Main text (Results and Discussion section):

“Finally, to evaluate the role of activated cross slip during creep, basal stacking fault energy (SFE) of $1/3 \langle 1\bar{1}00 \rangle_{\alpha}$ partial dislocations in pure-Mg, Mg-Nd and Mg-Nd-Zn were computed from their generalized stacking fault energy (GSFE) curves (see supplementary section). In Fig.5a the SFE for the intrinsic I2 stacking fault in *hcp* metals [43,44] correspond to the first minima of the GSFE curves, and they vary as $I2_{\text{Mg-Nd-Zn}} < I2_{\text{Mg-Nd}} < I2_{\text{Pure-Mg}}$. Note that $I2_{\text{Mg-Zn}} \approx I2_{\text{Pure-Mg}}$ (see extended Fig E9). For partials (with screw character [38]) to cross-slip to a different plane, they first need recombine into a perfect lattice dislocation, and the work done to drive that process is equivalent to dislocation dissociation energy of the lattice dislocation. The SFE was related to the dissociation energy (supplementary section). The calculated values are plotted as bars in Fig. 5b, which also compares the dissociation energies of pure-Mg, Mg-Nd and Mg-Nd-Zn with their *oop* and *ip* migration barriers (Fig. 3c-d). We note that addition of Zn to Mg-Nd increases the dissociation energy; meaning Zn is more effective in hindering activated cross slip in Mg-Nd-Zn alloys. Furthermore, the calculated higher *oop* migration energy (see Figure 5b) demonstrates that out-of-plane dislocation climb faces a greater energy penalty than *ip* vacancy diffusion and activated cross-slip. Similar results were also obtained for Mg-La and Mg-La-Zn (supplementary section and extended Fig. E9).”

Supplementary section

“**Calculation of stacking fault and dislocation dissociation energies.** To calculate stacking fault energies for various compositions we adopted the procedure previously described by Salloom et.al [8]. This involved the creation of 96 atom orthogonal cells (or 12 layers with 8 atoms per layer) with vacuum at both ends (Figure 5a). Furthermore, the cells were oriented with $[0001]_{\alpha}$ along the longer z axis, while the shearing xy plane (corresponding to $(0002)_{\alpha}$) was bound by two $[1\bar{1}00]_{\alpha}$ and $[11\bar{2}0]_{\alpha}$ orthogonal crystallographic directions. Faults were generated by shearing the top half of the cell along $[1\bar{1}00]_{\alpha}$ for $\sim 0.28\text{nm}$. This deformation creates an intrinsic I2 type stacking fault commonly seen in *hcp* structures [9,10]. The sheared cells were then energy minimized by allowing ionic relaxation only along the z axis. The SFE was calculated using the equation,

$$\text{SFE} = (\mathbf{E}_{\text{fault}} - \mathbf{E}_0)/A \quad (1),$$

where, E_{fault} and E_0 are the energies of the faulted and perfect structures respectively, and A is the interfacial area between the top and bottom slabs. In the calculations, all the solute atoms were placed in the plane of shear. For the ternary Mg-Nd-Zn and Mg-La-Zn systems, the Zn atom was placed as the first nearest neighbor of Nd/La.

For activated cross-slip to occur during creep, the partials must first recombine into a lattice screw dislocation, which then cross-slips to another slip plane [9,10,11]. Furthermore, the energy needed to drive during the recombination process will be equivalent to the dislocation dissociation energy (E_d), which can be evaluated using [11]:

$$E_d = \sigma_{r\theta}(d_{\text{partial}} b) d_{\text{partial}} \quad (2),$$

where, $\sigma_{r\theta}$ is the stress around a screw dislocation in polar coordinates, while d_{partial} and b (corresponding to $1/3\langle 1\bar{1}00 \rangle$) are the distance between the two partials and the Burgers vector, respectively. Equation (2) quantifies the energy required to recombine two partials separated by a distance of d_{partial} by applying stress $\sigma_{r\theta}(d_{\text{partial}} b)$ on the screw-partial. The Burgers vector was determined using the lattice parameter of Mg, while d_{partial} and $\sigma_{r\theta}$ were calculated using the expressions [10]:

$$\sigma_{r\theta} = \frac{G b}{\pi d_{\text{partial}}} \quad (3),$$

$$d_{\text{partial}} = \frac{G b}{8 \pi \text{SFE}} \frac{2-\nu}{1-\nu} \frac{2-3\nu}{2-\nu} \quad (4),$$

where $\nu (=1/3)$ is the Poisson's ratio, and SFE was obtained from DFT calculations. Thus, using equations (1)-(4) and generalized stacking fault energy (GSFE) curves, dislocation dissociation energy was calculated for Pure-Mg, Mg-Nd, Mg-La, Mg-Zn, Mg-Nd-Zn and Mg-La-Zn (see Figures 5 and E9)."

New Figure 5:

Figure 5 | Generalized stacking fault (GSF) and dislocation dissociation energies. (a) Plot depicting the variation of fault energy with planar displacement along $\langle 11\bar{2}0 \rangle_{\alpha}$ in Pure-Mg, Mg-Nd and Mg-Nd-Zn. Bottom inset figures shows two views of the orthogonal cell used in the calculations. Stacking fault energy correspond to the first minima of the GSF energy curves. Addition of Zn to Mg-Nd further reduces SFE by $\sim 17\%$. (b) Histogram comparing the energy required to dissociate a perfect $(1/2)\langle 11\bar{2}0 \rangle_{\alpha}$ dislocation into two $(1/3)\langle 1\bar{1}00 \rangle_{\alpha}$ partials for pure-Mg, Mg-Nd and Mg-Nd-Zn alloys. Adding Zn to Mg-Nd increases the dissociation energy by $\sim 20\%$. In-plane and out-of-plane vacancy migration energies, also indicated in the same plot, are significantly larger than the dissociation energies.

Based the above results we added the following to the conclusion section: “(4) Finally, Zn addition also likely hinders the activated cross-slip by reducing the stacking fault energies.”

Grain boundary sliding (GBS): GBS is a significant mode of deformation when grain sizes are of the order of 10nm or smaller. The alloy used in our study were prepared with high pressure die cast process. This process results in large grains and large intermetallic interdendritic solidification products (shown the extended Figure E1), which will likely retard grain boundary sliding (if any). Accordingly we have included the following statement in the supplementary section – “Past studies on creep tested microstructures (specifically at minimum creep rate) of Mg-Nd-La and Mg-Nd-La-Zn did not reveal any signature of grain boundary sliding [12,13]. It is suspected that the large intermetallic solidification products present at the interdendritic regions may have restricted grain boundary sliding.”

Comment 3. La is not discussed or studied at all in this work.

Response: The authors thank the referee for bringing up this issue. The supplementary section of the original manuscript mentions that La was added to improve the fluidity of the melt.

Per referee's comment, we have now elaborated on the role of La in the introductory and supplementary sections of the revised manuscript.

In the introductory section:

“Past work on Mg-La and Mg-Nd-La alloys have demonstrated that addition of La largely improves alloy castability [24], but minimally affects the creep behavior. This was attributed to the presence of La primarily inside the large interdendritic solidification phase rather than in Mg matrix [19,24,25] (also see supplementary figure S1). Thus, the remainder of the article will refer to alloys with and without Zn as Mg-Nd and Mg-Nd-Zn based. Notwithstanding, we have rigorously evaluated the role of La on the creep deformation behavior using DFT calculations in the supplementary section.”

Supplementary section: We made the following additions to the text and cited additional papers.

“**Role of La on creep deformation.** Past studies have indicated that La solutes improve castability, in comparison to Mg-Nd and Mg-Ce, by reducing hot tearing in the as-solidified products [1,30]. However, high pressure die cast (HPDC) Mg-La had the worst creep response of the three alloys [15], which has been attributed to the low solid solubility of La in the Mg matrix [14] (also see Table T2). Microstructure of HPDC Mg-La comprises of Mg matrix surrounded by a large interdendritic solidification phase skeleton [1,15]. Atom probe studies show that La preferentially partitions in such interdendritic phases in HPDC Mg-Nd-Zn alloys [12]. The same work also indicated very small composition of La (<0.01 at%) inside the Mg matrix [12]. Therefore, a possibility that La may influence the creep behavior mandated an assessment of La's influence in the present study.”

1. Easton, M., Gavras, S., Gibson, M., Zhu, S., Nie, J. F., & Abbott, T., Hot tearing in magnesium-rare Earth alloys. In *Magnesium Technology 2016* (pp. 123-128). Springer International Publishing, *and references there in*

12. Choudhuri, D., Jaeger, D., Gibson, M.A., & Banerjee, R., Role of Zn in enhancing the creep resistance of Mg-RE alloys, *Scr. Mater.*, 86, 32-35 (2014)

15. Zhu, S.M., Gibson, M.A., Easton, M.A. & Nie, J.F., The relationship between microstructure and creep resistance in die-cast magnesium-rare earth alloys, *Scr. Mater.*, 63, 698-703 (2010)

30. Easton, M. A., Gibson, M. A., Zhu, S., & Abbott, T. B., An a priori hot-tearing indicator applied to die-cast magnesium-rare earth alloys. *Metallurgical and Materials Transactions A*, 45(8), (2014)3586-3595.”

Based on referee's comments, we have conducted detailed DFT computations to understand how La solutes affect *vacancy migration pathways, stacking fault energies, and dislocation dissociation energies* in Mg-La and Mg-La-Zn. These results showed trends consistent with those obtained from Mg-Nd and Mg-Nd-Zn (present in the main manuscript). However, given the limited solubility of La in Mg matrix, it is unlikely that La will influence creep performance of Mg alloys. Details regarding the calculation of stacking fault energies and dislocation dissociation energies was provided in the previous section.

Due to word limit for the manuscript we have added the details in the supplementary section and created new extended figures.

“The role La was examined by calculating the minimum energy paths for vacancy diffusion (Figure E6) and GSFE curves of binary Mg-La and Mg-La-Zn alloys (Figure E9). The computed results indicated that vacancy migration barrier energies follow the trend: $E_{Mg-La-Zn}^{oop} > E_{Mg-La}^{oop} > E_{Mg}^{oop}$ (Figure E6a), and $E_{Mg-La-Zn}^{ip} \approx E_{Mg}^{ip} > E_{Mg-La}^{ip}$ (Figure E6b), which is similar to that seen in Mg-Nd and Mg-Nd-Zn alloys. Thus, out of plane vacancy diffusion (for dislocation climb) will be prevented when Zn is present near La atoms. Similar to Mg-Nd and Mg-Nd-Zn alloys, the SFEs in Mg-La and Mg-La-Zn alloys vary as $I_{Mg-La-Zn} < I_{Mg-La} < I_{Pure-Mg}$ (Figure E9a). Furthermore, Zn addition substantially increases the dislocation dissociation energy (Figure E9b); and hinder activated cross slip. Taken together we find the addition of Zn to Mg-rare earth alloys (or at least Nd and La) improves creep resistance by limiting dislocation climb and activated cross slip. However, the beneficial effects resulting from Zn-La interaction (similar to Zn-Nd interactions) will depend on the La content in the Mg matrix.

Figures summarizing DFT calculations of Mg-La and Mg-La-Zn systems are provided below.

New Extended Figure E6:

Figure E6. Energy vs. reaction coordinates plots for (a) out-of-plane and (b) in-plane vacancy migration in Mg-La and Mg-La-Zn.

New Extended Figure E9:

Figure E9. (a) plots comparing the generalized stacking fault energy (GSFE) curves of pure-Mg, Mg-Zn, Mg-La, and Mg-La-Zn. (b) Histogram comparing the energy required to dissociate a perfect $(1/2)\langle 11\bar{2}0 \rangle_\alpha$ dislocation into two $(1/3)\langle 1\bar{1}00 \rangle_\alpha$ partials in pure-Mg, Mg-La, and Mg-La-Zn are shown with in-plane and out-of-plane vacancy migration energies superimposed. The trends observed are consistent with Mg-Nd and Mg-Nd-Zn alloys.

Comment 4: Lines 22-23: No references are given as to which Mg alloys benefit from “small additions of amountsof elements such as Zn” with respect to creep resistance. Without the references, it is hard to assess the possible role of Zn on creep resistance.

Response: The authors thank the reviewer for this comment and have included the following in the manuscript introduction –

“Minor Zn addition to Mg alloys improve creep resistance of Mg-Gd-based alloys via the formation of new Zn-containing precipitate phases [18,22,23], and segregation of Zn and Gd atoms in as-quenched bulk and at the twin boundaries [21-23]. While clearly illustrating the effects of Zn, these studies also fundamentally suggest that Zn tend to occur near rare earth (RE) atoms, which also determines the precipitation and creep response.”

18. Nie, J. F., Precipitation and Hardening in magnesium alloys, Metall. Trans. A, 43A 3891 – 3939(1985)

21. Nie, J. F., Xiang Gao, & S. M. Zhu., Enhanced age hardening response and creep resistance of Mg–Gd alloys containing Zn, Scr. Mater., 53,1049-1053 (2005)

22. Nie, J. F., K. Oh-Ishi, Xiang Gao, & K. Hono., Solute segregation and precipitation in a creep-resistant Mg–Gd–Zn alloy, Acta. Mater., 56, 6061-6076 (2008)

23. Nie, J. F., Zhu, Y. M., Liu, J. Z., & Fang, X. Y., Periodic segregation of solute atoms in fully coherent twin boundaries. Science, 340,957-960(2013).

Comment 4: Lines 22-23: It should also be noted that there is nothing puzzling about this fact as the authors mention in the next sentence.

One of the objectives of this work was to understand how addition of Zn fundamentally changes the phase stability and vacancy diffusion in Mg-RE alloys. Such an insight is not accessible solely based on higher resolution experimental techniques. Therefore, **Our DFT work is essential to “bridge” the gap between experimental results and atomistic (and by extension bonding) underpinnings.**

Comment 5: Lines 45: It is well known that precipitation hardening with metastable phases is not a way to increase creep resistance as these precipitates would coarsen or transform to other metastable or stable phases during creep. Age hardening precipitates are rarely stable or even quasi stable at creep temperatures. It is noted that the authors have not shown TEM images from the tertiary creep stage (Fig. 1), where the creep rate increases; as such we cannot rule out precipitate coarsening and transformation as the main reason for the loss of creep resistance at longer life.

Response: The reviewer has correctly pointed out that precipitates can coarsen and/or transform to different phases during creep deformation, which inevitably reduced the creep resistance of the alloy. In fact we have documented such precipitate evolution in creep tested model Mg-Nd alloy microstructures [Choudhuri, D., Dendge, N., Nag, S., Gibson, M.A. & Banerjee, R., *Role of applied uniaxial stress during the creep testing on precipitation in Mg-Nd alloys*, *Mater. Sci. Eng. A.*, 612, 140-152 (2014)].

Regarding the current study, an order of reduction in the minimum creep rate of Mg-Nd-Zn (compared to Mg-Nd) alloy was previously attributed to the dynamic nucleation of γ'' in much higher number densities than those in Mg-Nd (primarily GP and β') [Choudhuri, D., Jaeger, D., Gibson, M.A., & Banerjee, R., *Role of Zn in enhancing the creep resistance of Mg-RE alloys*, *Scr. Mater.*, 86, 32-35 (2014)].

Our earlier TEM study of Mg-Nd-Zn microstructure indicated that γ'' persists even after ~4800Hrs of creep testing albeit with a lower number density compared to the microstructure at the minimum creep rate [Choudhuri, D., Jaeger, D. L., Srivilliputhur, S., Gibson, M. A., & Banerjee, R. (2015). *Creep response of a Zn containing Mg-Nd-La alloy*, *Magnesium Technology 2015*, pp. 35-39, Springer International Publishing]. This matter was also addressed in the original version, where the TEM results were included in the supplementary section due to space/word limit constraints. This result was illustrated in the extended Figure E4 of our paper:

Figure E4. Microstructure of Mg-Nd-Zn alloy after 4800Hrs of creep testing: (a) bright-field TEM image showing the existence of fine scale basal γ'' (indicated with arrows) and (b) High-resolution TEM recorded along $[11\bar{2}0]_{\alpha}$, from the region depicted with a dotted box shows that γ'' precipitates share a coherent interface with the parent Mg-matrix

Per referee's comments, we have clarified this matter in the main manuscript as follows-

"We will also emphasize that precipitates in both alloys will coarsen with prolonged creep (see extended Figure E4 and ref #36), and that will inevitably reduce creep lifetime of both systems."

Note that our TEM results clearly demonstrate that the γ'' precipitates are stable at 450K (creep testing temperature) for ~4800Hrs.

We would also like to point out that Figure 1 includes the microstructures of both alloys at their respective minimum creep rate conditions only, to highlight the differences in the precipitate phase (due to minor Zn addition) and their number densities. Importantly, minimum creep rates are commonly used to criterion to determine the governing deformation mechanism.

Comment 6: Lines 55 onwards: The simple way to determine the creep mechanism in an alloy is by generating Arrhenius plots with multiple temperatures with constant stress or multiple stresses at constant temperature to obtain the activation energy or the stress exponent. 600% or orders of magnitude increase in creep is very common in Mg alloys and these are of course not explained by mechanisms related to increase of strength which is quite different from creep resistance. The creep resistance of the alloys are not related to their strength or even to their high temperature strength because for the former, time is not a factor and the stresses involved are athermal versus thermally activated.

Response: The authors thank the reviewer for pointing out the relevance of "creep resistance", and apologize for the confusion that detracts from the *central message* of the role of minor Zn addition on the dislocation climb, precipitate phase stability, and activated cross slip during creep deformation and creep lifetime. The revised manuscript, specifically in the results and discussion section, replaces the phrase "*creep resistance*" with "*creep lifetime*" to avoid this confusion.

Regarding the Arrhenius plots, we respectfully point out that the need to also assess the dislocation substructure to gain a physical understanding of the stress exponent and the activation energies [Zhu, S. M., Nie, J. F., Gibson, M. A., Easton, M. A., & Bakke, P. *Microstructure and creep behavior of high-pressure die-cast magnesium alloy AE44. Metal. Mater. Trans. A*, 43, 4137-4144 (2012)]. **This would involve extensive TEM-based dislocation analysis of multiple conditions, which is beyond the scope of our work but is an excellent topic for a separate study.**

Regardless, it worthwhile to compare our work with the results of Zhu et.al. (*Metal. Mater. Trans. A*, 43, 4137-4144 (2012)), where they found the evidence of $\langle c+a \rangle$ dislocation climb, activated cross slip of basal dislocations, and non-basal slip of $\langle a \rangle$ type dislocations. As a comparison, our DFT computation, see Figure 5, show that both activated cross slip and dislocation climb is energetically possible (with the former requiring less energy than the latter). We also clearly show that the energy penalty increases appreciably with Zn addition. Furthermore, only the incorporation of the dislocation climb model can help us explain the significant order of magnitude enhancement in the minimum creep rates of the two alloys. In other words both dislocation climb and activated cross slip can occur in this alloy during creep testing at 450K and 90MPa.

Comment 7: Lines 177-183: The argument in this section that the precipitates can only strengthen 3-fold does not at all apply to creep and hence the argument and the claim in italics are not acceptable.

Response:

We respectfully point out that precipitates present in high number densities (e.g. γ' in Mg-Nd-Zn) interact with partials and perfect dislocation during cross slip. In that sense, “precipitation hardening” likely plays a major role in creep. Admittedly, as presented in the paper, precipitation strengthening models does not account for precipitate / dislocation interactions during cross slip.

Nevertheless, the authors understand the reviewer’s concern on this matter because the strengthening models were developed based on “shearability” of precipitates lying on the glide planes of dislocations. Therefore, we have removed the italicized statement form the revised version. In the current version, we have retained a part of the analysis, while emphasizing/acknowledging that mechanisms like dislocation climb and activated cross slip needs to be considered.

Therefore, the revised text reads as “To explain an order of magnitude reduction in the minimum creep rates we further used precipitate number densities in Mg-Nd and Mg-Nd-Zn, measured from BFTEM images, as inputs to different strengthening models to estimate creep enhancements in Mg-Nd-Zn (supplementary section). Furthermore, our DFT calculations revealed that elastic moduli of γ' are comparable to pure Mg (see supplementary materials) and hence do not significantly strengthen Mg-matrix by the modulus effect [18,37-39]. Classical precipitation strengthening models [37-39] predict only a 3-fold increase in the critical resolved shear stress (CRSS) for the γ' fractions estimated in Mg-Zn-Nd (see supplementary materials). However, any interpretation based solely on the microstructure fails to explain the observed improvement in the creep lifetime in Mg-Nd-Zn. Reasonable explanations must also consider vacancy interaction with precipitates and solutes, and acceleration of dislocation climb over impeding precipitates, driven by enhanced vacancy diffusion at high homologous temperature, and activated cross-slipping of screw dislocation [10-13]. ”

Comment 8 (a) Line 59-66: Again in this section none of the alloy composition limits are given. We finally see in the additional section that the alloy has 0.8wt% Zn. Well Zn has about 2wt % binary solubility in Mg at room temperature which increases towards 6wt% at high temperatures. But the alloys also have 3.4wt%Nd and 2.2wt%La. The authors should give the solubility limits of each element in alpha-Mg matrix of these alloys. Without this information the reader does not know if Zn is in solid solution (if so how much) and how much is in the precipitates. Nd on the other hand has no binary solubility but again it is not known if in the multi-component alloy the solubility limits change.

Response: We have addressed this criticism per our response to referee comment #1. Furthermore, we have tabulated the solubilities of Nd, La and Zn in Mg in new Table T2 in the *supplementary section*:

Table T2. Solubility of elements in Mg

Element	Max. solid solubility, wt%(at%)	Eutectic temperature, °C (K)
La	0.23 (0.04)	612 (885)
Nd	3.63 (0.63)	552(825)
Zn	6.2 (2.4)	340 (613)

Comment 8 (b): Line 59-66: Studies have indicated that Zn in solid solution is unusually surface active in Mg which is surprising because the Wigner Seitz radii do not predict that Zn would be surface active in Mg. It has been observed that Zn actually segregates to grain boundaries and stays there tenaciously and does not easily homogenize into the Mg grain even when soaked at high temperatures. If so it is highly likely that Zn would also segregate to dislocations and create dislocation atmospheres which are not easily lost at creep temperatures; this could explain the increase in creep resistance when Zn is added.

Response: The referee is correct to point out that Zn does indeed segregate. In our our previous work we studied the distribution of Zn using atom probe tomography [*Choudhuri et.al. Scr. Mater.*, *86*, p. 32-35 (2014)]. The Nd, La, Zn and Ga ion maps within an atom probe tip prepared by focused ion milling process using Ga ions are shown in the figure below below. The Ga ions are known to delineate dislocation, subgrain boundaries, and other defects. [*K.A. Unocic, M.J. Mills, G.S. Daehn, J. Microsc.* *240*, pp. 227 (2010)]. **Consistent with the referee's comment, these ion maps reveal traces of Ga ions in the subgrain boundaries, and the ion maps also highlight that Zn is indeed present at those boundaries. However, note that Zn is also present inside the matrix in much higher densities compared to La. Also note that the atom probe analysis also captured a portion of the intermetallic solidification product, which is rich in La (also see response to comment# 3)**

While we agree that Zn segregation does occur during creep deformation, still this alone cannot explain the observed improvement in creep properties. In comment #2 we have discussed in detail all the possible mechanisms that can be influenced by Zn addition.

Comment 8 (c): Line 59-66: It is also likely that Zn would stick to other defects and interfaces in the alloy; Zn could enrich at the outer surface of the precipitates and increase their stability or change their habit plane.

Response: We agree with the reviewer's observation that Zn can indeed attach itself to the precipitate matrix interface, for example Zn segregation at the precipitate/matrix interfaces of Mg-Zn alloys have been reported. However, our atom probe (Figure 1e) and HRTEM (Figure 2d) results did not reveal Zn segregation at γ'' /Mg interfaces. Although such segregations may have occurred at the interfaces of the substantially larger γ precipitates. However, as reported earlier [Choudhuri *et.al. Scr. Mater.*, 86, p. 32-35 (2014), and Choudhuri *et.al. Magnesium Technology 2015*, pp. 35-39, Springer International Publishing(2015)], the number densities of such precipitates were significantly less than the γ'' precipitates.

Comment 8 (d): Line 59-66: If Zn interacts with vacancies it would also make it difficult for the GP zones to form, which need vacancies (unless if Zn itself acts as nuclei for the GPs)

Response: GP zones were not observed in the Zn containing alloy, which is consistent with the comment of referee#1.

Comment 9: Line 206: The DFT calculations on vacancy migration pathways are interesting. Again it is not clear what composition range is simulated with the 96 atom supercell. In creep deformation there are many other pathways for vacancy migration (grain boundaries, fault boundaries, dislocations). It is not clear how the vacancy barrier has been related to the creep rates in Fig. 1 without an indication of the operative creep mechanism.

Response: The DFT supercell size (96 atoms) had a nominal composition of ~1at%Nd/La for binary systems and ~1%Nd/La + 1at%Zn for the ternary alloy. While the composition ranges studied by DFT reasonably reflect experimental values for Zn and Nd concentrations, the La concentration accessed by DFT is significantly higher than corresponding experimental value. We believe that this does not change the conclusion of our work because our objective was not to simulate the entire alloy concentration, rather we wanted to investigate how Mg vacancy migration is influenced by the presence of Nd/La/Zn solute species. The fact that the range of influence of solutes extends to no more than their second nearest neighbors, as shown here by us, gives us confidence in our above assertion.

We have described our approach in detail in the supplementary section of the original and current version of our manuscripts. This was done to comply with the word limitation of the journal. To summarize, we calculated the migration barrier energies for vacancies moving within basal plane (ip) and between two neighboring basal planes (oop). The *oop* migration is necessary for dislocation climb. We further correlated the calculated energies, for each pathway, with the well know empirical equation of creep due to Bird, Mukherjee and Dorn (BMD):

$$\dot{\epsilon} = A' kTG \left(\frac{\sigma}{G}\right)^n e^{-\frac{(E_v^f + E_v^m)}{kT}}$$

The BMD equation relates the steady state or minimum strain rate ($\dot{\epsilon}$) to the vacancy formation (E_v^f) and migration (E_v^m) energies. Using the BMD equation as well as inputs from our DFT calculations, we can explain an order of magnitude improvement in the minimum creep rate of Mg-Nd-Zn alloy seen in experiments summarized in Tables T6 and T7 given in the supplementary section.

Comment 10: Line 245: very difficult to accept covalent bonding in the Mg matrix unless if the DFT is actually looking at the composition ranges where intermetallics exist. The increase of covalent bonding in the intermetallics may have some bearing for creep resistance. Increased covalent bonding would lead to a less ductile intermetallic; this however can have both beneficial and negative effects on creep resistance.

Response: The authors would like to clarify that that Zn addition does not change the bonding character of the entire Mg matrix. Electronic charge density plots show the effect is largely confined ranges of the order of first nearest neighbor distances. We have also clarified this matter in the introductory section of the manuscript – “These fundamental material characteristics were further related to microalloying induced local lattice-level pockets with covalent character (separated by regions with metallic bonding) inside the Mg-matrix”.

Instead, the bonding character is perturbed in the vicinity of the first nearest neighbors of Nd and Zn, leading to the creation of a covalent character in this proximity region of Zn. **As noted in the Figure 4, the covalent character, by definition, results from the localization of electrons within a spatially confined region between two of more atoms [Sutton, A. P. *Electronic structure of materials*. Clarendon Press,(1993)].** This localization increases the bond strength, which we unequivocally demonstrate via our stiffness calculations (Figure 4 a and 4b). We further point out that the trend in the out-of-plane stiffness ($K_{Mg-Nd-Zn}^{1NN} \approx K_{Mg-Nd-Zn}^{2NN} > K_{Mg-Nd} > K_{Mg}$) along the c-axis also correlates extremely with the out-of-plane vacancy migration energies ($E_{Mg-Nd-Zn}^{oop} > E_{Mg-Nd}^{oop} > E_{Mg}^{oop}$).

Comment 11: The authors should either address the points mentioned above and improve /complete the analysis and the background on creep , or prepare the manuscript as a DFT study of the interaction of Zn with the precipitates and two vacancy pathways in Mg-0.6Nd-0.4La (at %) alloys.

Response: The authors have done their best to respond to all comments, including conducting many additional DFT calculations. Please note that all the experimental and computational techniques needed to be intimately coupled to answer this important problem. We believe that without this approach, modeling or experiments themselves individually will be inadequate to answer these outstanding questions.

We hope our clarifications will satisfy the reviewer.

Reviewer #2 (Remarks to the Author):

Comment 1: In this work, increased vacancy diffusion barrier is shown to correlate with 25% increase in inter-planar bond stiffness and is identified as the key mechanism in the enhancement in high-temperature creep strength. Overall, through an integrated first-principles modeling and high-resolution TEM and APT characterization study, the “localized bond stiffening” via micro-alloying is demonstrated as a viable route to enhance creep strength (by up to 600%). The findings are novel, the approach is rigorous using state-of-the-art capabilities and work represents a significant new advance in the field of precipitation-hardened Mg-alloys.

Response: The authors appreciate these encouraging comments of reviewer#2 and thank her/him for the service.

Comment 2: Minor correction: Change “complimented” to “complemented” on pg 3

Response: This change has been incorporated and is highlighted in yellow in the revised version.

Reviewers' Comments:

Reviewer #1:

Remarks to the Author:

The authors have shown due diligence in answering most of the concerns indicated by the reviewer. There is only one points that needs more work; it is related to comment 8 and the authors' response to that comment (both given below). The reviewer's new comment is at the end.

"Comment 8 (a) Line 59-66: Again in this section none of the alloy composition limits are given. We finally see in the additional section that the alloy has 0.8wt% Zn. Well Zn has about 2wt % binary solubility in Mg at room temperature which increases towards 6wt% at high temperatures. But the alloys also have 3.4wt%Nd and 2.2wt%La. The authors should give the solubility limits of each element in alpha-Mg matrix of these alloys. Without this information the reader does not know if Zn is in solid solution (if so how much) and how much is in the precipitates. Nd on the other hand has no binary solubility but again it is not known if in the multi-component alloy the solubility limits change

Response: We have addressed this criticism per our response to referee comment #1. Furthermore, we have tabulated the solubilities of Nd, La and Zn in Mg in new Table T2 in the supplementary section:

Table T2. Solubility of elements in Mg.....

NEW COMMENT

The reviewer apologizes for not being clear about comment 8 in the first review. Perhaps the authors, having excellent expertise in computational materials science, are not equally well-versed in alloy phases. Most metallic alloys will have a primary (solid solution phase or pure metal) matrix phase and perhaps second phases (metallic or intermetallic). The metallic phases (pure or solid solution) have metallic bonding (majority of cases) and the intermetallics will have part metallic part covalent bonding (or only covalent). In this work we should be interested in the bonding in the solid solution alpha-Mg phase since we are interested in its creep deformation behavior. We must specify the composition limits of this primary solid solution phase because it is only in that phase that we would be interested in whether the Zn has metallic or covalent bonding. The authors have not specified the composition limits of the alpha-Mg phase. The solubility limits of the individual solutes do not help us in this instance, because in multicomponent alloys with many solutes, these limits will change. The authors need to do a thermodynamic calculation (e.g. Thermocalc or FactSage) and provide us the composition range of alpha-Mg. They would then need to do their first-principles calculations within that composition range (i.e simulate the alpha-Mg composition in that multi-component alloy), and then, if Zn has covalent bonding in that alpha-Mg phase composition that would make the conclusions and the claims valid. Otherwise, if Zn has covalent bonding when the alloy composition falls into the intermetallic phase well then it of course should; intermetallic phase have covalent bonding.

Second Phase Response to Reviewer #1 Comments

Referee #1 Comment: Most metallic alloys will have a primary (solid solution phase or pure metal) matrix phase and perhaps second phases (metallic or intermetallic). The metallic phases (pure or solid solution) have metallic bonding (majority of cases) and the intermetallics will have part metallic part covalent bonding (or only covalent). In this work we should be interested in the bonding in the solid solution alpha-Mg phase since we are interested in its creep deformation behavior. We must specify the composition limits of this primary solid solution phase because it is only in that phase that we would be interested in whether the Zn has metallic or covalent bonding. The authors have not specified the composition limits of the alpha-Mg phase. The solubility limits of the individual solutes do not help us in this instance, because in multicomponent alloys with many solutes, these limits will change. The authors need to do a thermodynamic calculation (e.g. Thermocalc or FactSage) and provide us the composition range of alpha-Mg. They would then need to do their first-principles calculations within that composition range (i.e simulate the alpha-Mg composition in that multi-component alloy), and then, if Zn has covalent bonding in that alpha-Mg phase composition that would make the conclusions and the claims valid. Otherwise, if Zn has covalent bonding when the alloy composition falls into the intermetallic phase well then it of course should; intermetallic phase have covalent bonding.

Response: Once again, we thank the reviewer for the thorough review and suggestions. We are also grateful to the reviewer for clarifying his question. Accordingly, we have analyzed our results from the perspective experimental results and DFT computations.

In the second round of review, the referee raised the following two critical questions:

1. What is the composition of Zn in the Mg-matrix or alpha Mg solid solution?
2. How the concentration of Zn in the Mg-matrix relates to the DFT computation?

The referee's concern was: "Otherwise, if Zn has covalent bonding when the alloy composition falls into the intermetallic phase well then it of course should; intermetallic phase have covalent bonding."

Regarding the question#1, the reviewer has suggested that we perform thermodynamic calculations using CALPHAD-based approaches (e.g. Thermocalc, FactStage, Pandat etc.) to determine the highest Zn solubility in α -Mg matrix.

While solution thermodynamic models of Mg-Nd-Zn do exist, the accuracy of predictions afforded by such models has not been conclusively established. These solution thermodynamic models must be accurately calibrated with experimental results (phase and composition analysis using SEM/EDX, TEM etc.) obtained from systematically heat treated specimens of various controlled compositions. Furthermore, our experience with such efforts in Mg-Nd [Choudhuri et.al. *J. Mater. Sci.*, 49, p. 6986 (2014)] and Ti-V [Choudhuri et.al. *Acta Materialia* 130, p. 215 (2017)] alloys suggest that metastable solid state pathways (e.g. early stages of precipitation in Mg-Nd and Mg-Nd-Zn systems, phase separation etc.) can be thermodynamically modeled

accurately only by coupling with high resolution experimental techniques like atom probe tomography (APT) for measuring compositions at nanoscale and TEM/HRTEM/HRSTEM to determine structure. When we compared our experimental results with the predictions from the solution thermodynamic models, using existing databases for the Mg-Nd-Zn system, the results were not in agreement in many cases. Undertaking the development of a new solution thermodynamic database and models for the Mg-Nd-Zn system requires tremendous effort spanning at least couple of years. Such a study proposed by the referee, while important, is beyond the scope of our current study.

Nevertheless, we agree with the referee on the importance of answering whether the Zn concentration used in our original DFT calculations lie within the solubility limit of α -Mg system. So, instead of using CALPHAD approach proposed by the referee, we will use high-resolution experimental characterization of Zn concentration in well-annealed Mg-matrix to answer this question.

Figure R1 shows the 3D reconstruction of an atom probe tip, which reveals a portion of the interdendritic solidification product (ISP) and the surrounding Mg matrix shown in supplementary Figure E1. Figure R1(a) shows the Nd, La and Zn ion map while Figure R1(b) shows a 1D concentration profile along the length of the atom probe tip. Note that ISP is rich in La (as well as Nd and Zn) while the regions away from the ISP are depleted in solutes (as expected). **Farther away, the solute concentration in α -Mg matrix is richer at ~ 0.2 at% Zn and Nd reaches as high as ~ 0.5 at%. The average concentration of the alloy itself was Mg-0.6Nd-0.6La-0.3Zn (at%). Importantly, measurements of at least 4 APT tips showed that Zn concentration in the Mg-matrix varied from 0.1-0.4at%.** Note that La had very negligible solubility in α -Mg matrix as discussed in the previous round of review.

Figure R1 (a) Ion maps showing the distribution of Nd (blue), La (red) and Zn (green) ions in the APT tip obtained from a creep tested Mg-Nd-Zn specimen. A cylindrical region of interest (ROI) is also shown in the Zn ion map. (b) 1D composition profile measured using the cylindrical ROI.

Regarding the question#2: The original manuscript used 100 atom supercells of α -Mg with 1 Zn and 1 Nd substitutional each. This composition corresponded to *1 at% for both Zn and Nd atoms*, values that are *about twice as large as the nominal, experimentally determined composition* of our alloy. This point, though mentioned in the original manuscript, was not adequately highlighted. **Therefore, based on referee's question, we used the APT composition data to construct two supercells with 500 atoms each and performed DFT calculations in these larger systems to determine the effect of Zn on bonding (Figure R2 below). The composition of Zn and Nd substitutionals in our new 500-atom supercells was 0.2at.% each, which is well within the experimentally measured compositions of Nd and Zn dissolved in α -Mg matrix.**

The Mg-0.2Nd-0.2 Zn(at%) (Figure R2(a)) and Mg-0.2Nd(at%) supercells (Figure R2(b)) were relaxed and their valence charge densities were computed. **These calculations consumed over 28,000 processor hours for the two 500 atom-systems. This substantially large time requirement prevents us from routinely using such large systems.**

The charge densities from our 500 atom supercells are shown as yellow isocharge surfaces in Figures R2(a) and R2(b). **Figures R2(a) and R2(b) show that valence electron distribution is perturbed near the valence solute environment, while the remainder of the Mg-matrix remains undisturbed.** As in the original manuscript, the Mg-Nd-Zn valence electrons are localized along the c-axis of α -Mg indicating local covalent character. **That these results are quantitatively consistent with our earlier findings in the 100-atom DFT supercell described in the main manuscript Figure 4 is not surprising.** This is because of the fact that the observed charge localization extends to only nearest neighbor plane of α -Mg along the c-axis, and the supercell lengths for both 100 and 500 atom cells are much larger than c-axis interplanar distance. *In summary, these results indicate that proximity of Zn to Nd will enhance covalent bond character by localizing charge density along the c-axis, but will not manifest when Nd and Zn are at distances beyond first nearest neighbor separation of α -Mg as shown in the original manuscript.* **Thus, the results and conclusions presented in the original manuscript remain unchanged.**

Accordingly, in the main manuscript we have included the following statement and highlighted in yellow:

“Such valence electron density localization was also verified in larger 500 supercells where Nd and Zn concentrations are well within their experimentally measured values.”

We finally point out that the electron localization along the c axis increases the bond stiffness and vacancy migration energies (as indicated in Figure 3), which impacts the creep behavior. *Note that it is not our intention to replicate the composition of the entire Mg-Nd-Zn-La alloy. Rather, the 100-atom supercells used in the manuscript with compositions Mg-1Nd-1Zn(at%) and Mg-1Nd(at%), and shown in Figure 3, simply modelled the local environment of the solutes to elucidate how they affect bonding and diffusion, and ultimately understand their role in enhancing creep life of these alloys.*

We apologize for the confusion on this matter in our earlier manuscript. *Our intention was to highlight our discovery that small pocket regions with covalent character exists as small perturbations around solute environments* (specifically Nd-Zn in Figure R2(a)). **This does not drastically alter the metallic bonding character of the remaining α -Mg matrix** (Figure R2b).

Figure R2. DFT computations of valence electron charge densities in 500 atom supercells of (a) Mg-0.2at.%Nd-0.2at.%Zn and (b) Mg-0.2at.%Nd

Reviewers' Comments:

Reviewer #1:

Remarks to the Author:

The authors have made a good effort to respond to the concerns of this reviewer. The responses, however, show that the manuscript cannot be published in the form submitted and requires major rewrite with a different focus.

1)The beginning of the paper should describe briefly the alloy composition, the phases (alpha-Mg so lid-solution matrix and the second phases) and give the chemical compositions of these phases. In this respect, it was rather easy to use FactSage (under Scheil cooling conditions) to assess the phases in the alloy and their compositions but experimental determination is also excellent, especially if it can show that Zn actually enriches in certain regions (it can be hypothesized that these regions are likely, grain boundaries, interdendritic regions, interfaces, dislocations) of the Mg matrix phase. One such indication has been reported by

A. Becerra, M. Pekguleryuz, "Effects of Lithium, Indium and Zinc on the Grain Size of Magnesium", J. Mater. Res., v. 24, n. 5, 2009, pp. 1722-1729

2)The authors can then show that when Zn concentrates in these regions it can exhibit covalent bonding.

3)Having shown all this, the authors need to explain/postulate what kind of effect covalently bonded Zn can have on the creep behaviour of this alloy under the stress and temperature ranges tested. Does it change activation energy for vacancy creation? Does it lower dislocation mobility? How are these related to bond stiffness (which governs elastic deformation rather than plastic deformation or creep)?

4)The title of the manuscript must change and cannot be used as is; it should reflect what the study can conclude and not what it postulates.

Round-3 Rebuttal to Referee #1

We thank the Referee #1 and the associate editor for their service. **We are delighted that both the referee and the editorial team agree that our manuscript (MS) is now technically sound.** This improvement is the outcome of two rounds of referee questions, which lead to new DFT simulations, experimental observations, and detailed analyses.

Our creep experiments showed an unexpected and remarkable enhancement of creep life in Mg-Nd alloys when they were micro-alloyed with Zn. This extraordinary behavior provided impetus to use extensive compositional and structural characterization (involving Atom Probe, and both conventional HR-S/TEM) and complementary DFT calculations to decipher the underlying, fundamental atomistic mechanisms. These have been described extensively in our MS and past rebuttal documents.

We are not conducting alloy design and neither are our experimental results postulates. Rather, we systematically investigated and presented new atomistic mechanisms behind the extraordinary creep resistance of Mg-RE alloys. **Thus, we are rather perplexed by the most recent referee comments (listed below) since these specific comments have already been addressed, in great detail, in our revised manuscript and supplementary material via discussions, figures, and tables, in response to rounds 1 and 2 of rebuttals.**

Furthermore, to add to our confusion, the referee explicitly states that we have adequately addressed her/his technical questions in both rounds 1 and 2 but then wants us to extensively rewrite this paper. Our response to rounds 1 and 2 of the rebuttal would clearly show that we seriously and willingly addressed many questions regarding the technical correctness in our work. However, this comment is subjective and unique styles of different authors must be respected.

Below, we address each referee comment.

Reviewer #1 (Remarks to the Author):

Referee: The authors have made a good effort to respond to the concerns of this reviewer.

Our Response: We are delighted that the referee recognizes our efforts.

Referee: The responses, however, show that the manuscript cannot be published in the form submitted and requires major rewrite with a different focus.

Our Response: We are at loss here. In the single question in round-2 of refereeing, the referee questioned whether the composition we used was for a random alloy or an intermetallic. Specifically, the referee wanted us to make sure no intermetallics formed at the composition we considered in the MS. We showed experimental results (Atom probe Data) indicating that we had a random solid solution for the composition used in creep experiments. We subsequently carried out simulations with a 500-atom DFT supercell whose composition matched experiments and showed conclusively that our results are unchanged. Thus, we do not understand the basis of this confusing comment.

Referee: 1) The beginning of the paper should describe briefly the alloy composition, the phases (alpha-Mg so lid-solution matrix and the second phases) and give the chemical compositions of these phases.

Our Response: This was done in the revision-1 of our MS and supplementary material after first round of rebuttal.

Referee: In this respect, **it was rather easy to use FactSage (under Scheil cooling conditions) to assess the phases in the alloy and their compositions but experimental determination is also excellent**, especially if it can show that Zn actually enriches in certain regions (it can be hypothesized that these regions are likely, grain boundaries, interdendritic regions, interfaces, dislocations) of the Mg matrix phase. One such indication has been reported by

A. Becerra, M. Pekguleryuz, "Effects of Lithium, Indium and Zinc on the Grain Size of Magnesium", J. Mater. Res., v. 24, n. 5, 2009, pp. 1722-1729

Our Response: In our round-2rebuttal, we explained that since we have the experimental data on the phases present and their compositions (based on TEM and 3D atom probe results) we don't need to use solution thermodynamic (CALPHAD) models to determine these results. We also pointed out that it may not always be a good idea to just use CALPHAD-based models (FactSage, ThermoCalc, PANDAT) since we have encountered situations where the lack experimentally validated databases for these models, leads to erroneous predictions. Additionally, the above-mentioned paper studies 2-component systems. But, we have rare-earth elements in our system in addition to Mg and Zn. Thus, we have to conduct extensive validation of the solution models used in CALPHAD. This was clearly given as a justification for not using CALPHAD as suggested by the referee, and for using experimental data like we did.

Referee: 2) The authors can then show that when Zn concentrates in these regions it can exhibit covalent bonding.

Our Response: Our rebuttal (referring to previous papers), MS and supplementary material in many places, figures, and table shows that Zn is located in the alloy matrix.

Referee: 3) Having shown all this, the authors need to explain/postulate what kind of effect covalently bonded Zn can have on the creep behavior of this alloy under the stress and temperature ranges tested. Does it change activation energy for vacancy creation? Does it lower dislocation mobility? How are these related to bond stiffness (which governs elastic deformation rather than plastic deformation or creep)?

Our Response: All the questions listed by the referee have already been clearly addressed, in explicit detail, in our revised MS and supplementary materials at numerous places, figures, and tables. Our DFT calculations on how Zn and Nd affect vacancy formation and migration energies have been summarized in figures and discussions in this paper. Calculations and figures also show how Zn influences in-plane and out-of-plane bond-stiffness. Thus, again these comments are rather perplexing.

Referee: 4) The title of the manuscript must change and cannot be used as is; it should reflect what the study can conclude and not what it postulates.

Our Response: We respectfully disagree with the referee. There is no postulate in our paper. Our DFT calculations provide the fundamental mechanisms that rationalize the experimentally observed enhancement of creep in Mg-RE alloys. Nevertheless, for the sake of enhanced clarity, we have modified our title to, “**Exceptional increase in the creep life of magnesium rare-earth alloys due to localized bond stiffening**”. This is highlighted in yellow in the revised manuscript.

Reviewers' Comments:

Reviewer #3:

Remarks to the Author:

Upon careful reading of the paper and of the authors' response to earlier reviewers it is my opinion that this paper deserves publication in Nature Communications. In particular, it is my contention that all the points raised by reviewer #1 (round 3) have been convincingly addressed in their revised manuscript.

The authors have used a complex approach, including first principles calculations and advanced characterization techniques, such as APT and high resolution TEM, to examine to a great detail the microstructure of the Mg-Nd and Mg-Nd-Zn alloys and to explain the reason underlying the dramatic increase in the creep life of the ternary alloy.

In my opinion, they convincingly demonstrate that the addition of Zn leads to an enhanced localized bond stiffness that is a major contributor to creep strength (among other factors, that are also described in the paper).

I have no doubt that this work will open new avenues for the development of creep resistant Mg alloys.